# Simulated Cropping Season Effects on N Mineralization from Accumulated No-Till Crop Residues

Rashad S. Alghamdi [1,*], Larry Cihacek [1] and Qian Wen [2]

1. Department of Soil Science, North Dakota State University, Fargo, ND 58108, USA; larry.cihacek@ndsu.edu
2. Department of Statistics, North Dakota State University, Fargo, ND 58108, USA; qian.wen@ndus.edu
* Correspondence: rashad.alghamdi@ndsu.edu

**Abstract:** The adoption of no-till management practices has increased in the United States over the last decade. In the state of North Dakota, approximately 5.7 million hectares of cropland is managed under no-till or conservation tillage management practices. Even though conservation tillage is known for building soil health, increasing soil organic matter, capturing soil moisture, and reducing wind and water erosion, it also presents a unique best management practice since an increased mass of crop residue remains on the soil surface. Producers are concerned about whether plant needs are being met by nitrogen fertilizer that is currently being applied based on current North Dakota recommendations for long-term no-till systems. A Forman clay loam soil (fine-loamy, mixed, superactive, frigid Calcic Argiudolls) was used in this study, as it represented glacial till soils of the region. We examined whether N mineralization from surface-applied crop residue would result in similar or different results when compared to crop residue mixed into the soil. Soil freeze-thaw contribution to soil N mineralization was also evaluated. Six residue treatments with different C/N ratios including corn (*Zea mays* L.), soybean (*Glycine max* L.), forage radish (*Raphanus sativus* L.), winter pea (*Pisum sativum* L.), spring wheat (*Triticum aestivum* L.), and winter wheat (*Triticum aestivum* L.) were used. Five 10–14-week cycles with a three-week freeze period between each cycle at 0 °C were evaluated for NO$_3$-N production. Crop residues with a narrow C/N ratio contributed to greater instances of N mineralization during each incubation cycle, and the accumulation of crop residues with a wide C/N ratio over each incubation cycle following the first incubation did not offset the immobilization trends observed in the first incubation. A change in N mineralized in the untreated control soil during the last two incubation cycles may have been caused by freeze-thaw effects or a shift in microbial population due to a lack of fresh C inputs.

**Keywords:** N mineralization; C/N ratio; crop residue; N availability

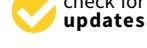



## 1. Introduction

In a no-till agricultural system, crop residue remains on the soil surface, whereby in a drier environment with lower soil temperatures may slow and/or reduce nitrogen mineralization, with the potential to affect crop yields. In areas where there is a short growing season, no-till practices have been perceived by producers to delay planting and slow crop emergence as a result of delayed soil warming and drying [1–4]. In a frigid environment, such as that in the Northern Great Plains, these concerns are more pronounced, as crop residue tends to accumulate when high residue crops (corn, small grains) are left on the soil surface as a result of no-till practices. Alghamdi et al. [5] examined soil warming and drying in a frigid environment for corn-soybean systems and provided evidence to suggest that these perceived delays are not related to moisture and temperature of the soil. Daigh et al. [6] have also reported research on full-production scale farms in the Red River Valley of Minnesota and North Dakota (i.e., frigid environment), finding that producer perceptions of delayed warming and drying of the soil do not translate into yield losses. This begs the question of where these perceived losses are coming from.

Alghamdi et al. [7] concluded that any differences observed in yield were attributed to fertilizer application methods and rates. Many earlier studies concluded that an addition of nitrogen fertilizer may be necessary under a no-till system to increase nitrogen (N) in the deeper soil profile [8–10], yet modern recommendations often overlook previous decades of research [11]. The rate of soil N mineralization is dependent upon crop residue factors that include, but are not limited to, the type and quantity of residue and other soil factors, such as temperature and moisture in the residue environment and organic matter (OM) in the soil [12]. As residue continues to accumulate on the soil surface in a no-till system, it is necessary to determine if enough N is being provided to subsequent crops to offset the potential immobilization in the system. Schoenau and Campbell [13] reported that conservation tillage such as reduced and no-till systems results in greater surface accumulation of crop residue on top of the soil surface, which slows residue decomposition of wide C/N ratio crops. No-till systems provide added benefits of increased soil moisture and organic matter, although reduced availability of oxygen from aeration needed by microbes may slow the mineralization of nitrogen from the crop residue. In addition to increasing soil moisture, no-till systems can be susceptible to water saturation as a result of increased crop residue on the soil surface. Thus, the activity of microbes may be reduced along with soil shading and slower soil drying due to cooler temperatures [14]. A laboratory study on North Dakotan soils concluded that the type of crop residue and organic matter present may increase N immobilization when crop residue has a wide C/N ratio (>25:1) as compared to crop residue with a narrow C/N ratio (<25:1) [7]. Green and Blackmer [15] examined the rate of N fertilization and its effect on residue decomposition on corn–corn and corn–soybean fields in Iowa. They determined that N immobilization tended to decrease with increased N fertilization as a result of the fertilizer contributing N to the residue biomass. Increasing rates of N fertilization decreased the time required for N mineralization to occur. This process was expedited when soybean residue was the proceeding crop due to the nature of the soybean residue. Li et al. [16] found that N mineralization occurred more quickly with crop residue placed on the soil surface versus crop residues incorporated into the soil. Satchell [17] explained the biological process of decomposition, where plant tissues are broken down by microorganisms. These microorganisms can break down residue more quickly when plant residue is incorporated with the soil [18] as compared to remaining on the soil surface [19]. Coppens et al. [20] found that incorporated crop residues decomposed faster than leaving residue on the soil surface and that higher N fluxes were more pronounced with crop residue incorporation. For the residue that did remain on the soil surface, the absence of moisture was a greater limitation to decomposition than N itself. Often, to aid in the breakdown of high lignin content crop residue, such as corn, many producers apply liquid N to their fields after harvest. Al-Kaisi et al. [21] examined corn residue decomposition and the application rate of liquid urea ammonium nitrate (UAN) following harvest and concluded that increased rates of decomposition were not attributed directly to the application rates of N, but to increased air and soil temperature. They recommended the limiting of liquid N application under less-than-ideal air temperature conditions and/or limited availability of soil moisture during critical periods of residue decomposition. Additional studies have determined similar finding where decomposition effects were attributed to temperature and moisture availability [22–24]. In cooler climates of the Northern Great Plains, such as in North Dakota, highly variable precipitation and temperature conditions during crucial periods of crop N needs can lead to the rate of residue decomposition being less predictable, resulting in less certain N mineralization for crop needs. Vigil and Kissel [25] examined the impact of temperature on nitrogen mineralization and residue decomposition in Kansas soils. Their findings indicated that higher incubation temperatures resulted in increased mineralization. At temperatures lower than 35 °C, microbial activity decreased. Nitrogen from sorghum and soybean residues mineralized faster at 35 °C than 25 °C, 15 °C, and 5 °C, respectively. Aher et al. [26] examined C/N ratios and the mass of residue on the soil surface near Forman, ND, and indicated potential N deficits to succeeding crops

ranging from 56 to 105 kg N ha$^{-1}$ following winter weathering of the crop residue. North Dakota has a frigid climate where residue decomposition and nutrient mineralization from crop residues can be limited by the short frost-free period of 100–135 days [27,28]. In the Canadian prairie, an incubation study was conducted to examine the factors influencing the stability of microbes in the soil [29]. Results indicated that freezing and thawing periods had a strong effect on releasing N and other nutrients back into the soil when compared with wetting and drying cycles. The reason for this was that the freeze and thaw periods enabled microorganisms to feed on decomposed residue. During freezing and thawing, soil aggregates are disturbed, exposing sheltered residue to microbes that help release nutrients back to the soil [30,31] or may break up residue particles to provide a greater residue surface area for greater microbial access. During the winter, native soil organic matter mineralizes to release N [32] due to microorganism abate and releases amino acids and simple sugars back into the soil [33]. During thaw periods, surviving microorganisms are active and feed on the nutrients released from dead microorganisms [34]. As a result, many studies have noted the increase in microbial respiration following freezing and thawing periods [33–36]. Current recommendations for N management in North Dakota do not regard temporal residue decomposition and N mineralization and immobilization effects that vary with different types of crop residue in their recommendations. Instead, standardized recommendations are cited for long-term no-till systems (i.e., managed for six or more years) where producers are to take a 34 kg N ha$^{-1}$ credit [37]. Over the last ten years, recommendations for North Dakota have varied. Standardized N recommendations are useful in warmer and moister climates, which allow for optimum N mineralization from crop residue. Management recommendations for nitrogen fertilization of soil should aim to address concerns of the producer and advance best management practices. The objectives of this study are: (1) evaluate if the type and quantity of crop residue are offsetting the N immobilization, (2) evaluate the influence of freezing and thawing on crop residue decomposition, and (3) evaluate changes in N mineralization due to repeated residue addition over several simulated growing seasons.

## 2. Materials and Methods

### 2.1. Experimental Design

A laboratory study using a randomized complete block design with three replicates was set up using one soil and six residue treatments, plus an untreated soil control (*n* = 21). The soil was a Forman soil (fine-loamy, mixed, superactive, frigid Calcic Argiudolls) [38], similar to many soils of glaciated parts of the region. The six residue treatments were corn (CR), soybean (S), forage radish (R), pea (P), spring wheat (SW), and winter wheat (WW) (Table 1). Fresh individual crop residues were collected immediately in the fall following harvest and analyzed for carbon (C) and nitrogen using an Elementar Vario Max® CN analyzer (Ronkonkoma, New York, NY, USA) during a previous study conducted by Aher et al. [26]. Upon collection, the residue was oven-dried at 60 °C and ground in a Wiley mill to pass a <2 mm screen. Bulk Forman soil was collected at the Conservation Cropping System Project site [26] near Forman, North Dakota (97°38′38″ N 46°05′05″ W). The bulk soil sample was air-dried, crushed, and sieved through a 2 mm screen. Residue treatments (*n* = 18) were prepared containing 15 g of soil mixed with 15 g of quartz sand (20 mesh) with 0.50 g of residue placed on the soil surface following the procedure of Stanford and Smith [39]. The quantity of residue represents the equivalent of 6.25 Mg/ha in a field environment [7]. Soil only (*n* = 3) contained 15 g of soil mixed with 15 g of sand. Samples were then transferred to labeled glass leaching tubes where the soil was placed into the tube, and crop residue was then placed on top of the soil surface to simulate no-till residue accumulation. Leaching tubes were kept in a constant temperature room at 22 °C simulating average soil temperatures in the region during the growing season and incubated as described by Stanford and Smith [39]. Before the soil samples were transferred to the leaching tubes, a small amount of glass wool was placed at the bottom of the leaching

tubes to prevent soil sediment loss during leaching and was placed at the top to prevent soil and residue disturbance during leaching solution application.

**Table 1.** Six crop residue treatments used for the five incubation cycles, their scientific name, abbreviation, and C/N ratios.

| Residue Treatment | Scientific Name | Abbreviation | C/N Ratio [1] |
|---|---|---|---|
| Corn | *Zea mays* L. | CR | 73 |
| Soybean | *Glycine max* L. | S | 53 |
| Forage radish | *Raphanus sativus* L. | R | 8 |
| Winter pea | *Pisum sativum* L. | P | 18 |
| Spring wheat | *Triticum aestivum* L. | SW | 76 |
| Winter wheat | *Triticum aestivum* L. | WW | 101 |

[1] C/N ratios for corn, soybean, forage radish, winter pea, spring wheat, and winter wheat were determined using near-infrared reflectance (NIR).

Five incubation cycles and four freeze–thaw periods were conducted for the experiment. Each cycle represented an annual growing season. Between each cycle, the soil treatments were frozen for three weeks to represent an annual freeze–thaw cycle. Following each freeze ($n = 4$), an additional 0.5 g of individual crop residue was placed on the soil surface (total = 2.5 g per leaching tube over the course of the study). Biweekly leachings were conducted at 2, 4, 6, 8, and 10 weeks for the first cycle. For the second cycle, additional leaching was conducted at 12 weeks. For cycles three, four, and five, additional leachings were conducted at 12 and 14 weeks to determine the potential of cumulative effects of the simulated growing seasons. For the first leaching of the samples and due to the air-dry nature of the soil/sand/residue column, 50 mL of 0.01 M $CaCl_2$ was added to the glass tubes in 10 mL increments, followed by 10 mL of nutrient solution as described by Stanford and Smith [39] to remove ambient levels of $NH_4^+$-N and $NO_3$-N. Subsequent leachings used 30 mL of 0.01 M $CaCl_2$ and 10 mL nutrient solution. Parafilm® (Menasha, WI, USA) was used to cover the incubation tubes between leachings to prevent contamination and preserve soil water content during this study. Air vents were created in the film to allow for soil respiration. At the end of an incubation period, the leaching tubes were then transferred into the freezer for three weeks at 0 °C. Leachate was collected, covered, and refrigerated (8–24 h) if necessary until mineral N ($NO_3$-N and $NH_4$-N) analysis was conducted using a Timberline TL 2900 $NH_4$/$NO_3$ analyzer (Timberline Instruments Inc., Boulder, CO, USA). Results are adjusted for soil mass and reported as mg N $kg^{-1}$ soil in order to relate our results to quantities that might be experienced in field environments.

### 2.2. Statistical Analysis

A repeated-measures analysis of variance (ANOVA) was used to determine the effects of crop residue treatment, cycle, incubation period, and their interactions on $NO_3$-N mineralization. As the measurements of $NO_3$-N mineralization collected from the same experimental tube unit are related over time, the model imposed a covariance structure on the error term of the model. Akaike's information criteria (AIC) was used to determine the appropriate covariance structure, and the smaller the AIC value, the better. Throughout this paper, AR(1) covariance structure was used due to it always producing the smallest AIC value. The least square mean (LS mean) of each level of the freeze and thaw periods was estimated, and the significance of the difference between all possible pairs of these LS means was identified with Tukey's honest significant difference (HSD) at a significance level of 0.05. In addition, LS mean of each individual crop residue treatment within each freeze and thaw cycle was estimated, and an HSD test with a significance level of 0.05 was performed to test the significant difference among all possible pairs of the LS means. Moreover, LS mean of each individual crop residue treatment for each incubation period within each freeze and thaw cycle was estimated, and an HSD test with a significance level of 0.05 was used to find the LS means that are significantly different from each other. In order to detect

the change of $NO_3$-N mineralization for the bare, unamended soil over incubation cycles for each freeze and thaw period, a simple one-way ANOVA with a repeated measurements model was fitted. LS mean of each incubation period within each freeze and thaw cycle was estimated, and an HSD test was used to find the LS means that are significantly different from each other. All analyses were conducted using SAS version 9.4 [40].

## 3. Results and Discussion

Mean soil $NO_3$-N mineralization capacity for individual crop residue treatments varies among freeze and thaw periods from 1.88 mg $NO_3$-N kg$^{-1}$ to 58.97 mg $NO_3$-N kg$^{-1}$ (Table S1, Supplementary Materials). The mean soil $NO_3$-N mineralization capacity (range in means) peaks during cycle 3 while displaying the narrowest range in mean soil $NO_3$-N mineralization capacity in cycle 5. Freeze and thaw period means varied from 4.95 mg $NO_3$-N kg$^{-1}$ to 17.84 mg $NO_3$-N kg$^{-1}$, with cycles 4 and 5 significantly higher than cycles 1, 2, and 3. For all freeze and thaw periods, the control (bare, unamended soil) means show net N mineralization varying from 1.74 mg $NO_3$-N kg$^{-1}$ to 22.72 mg $NO_3$-N kg$^{-1}$. Both cycle 4 and cycle 5 display higher mean soil $NO_3$-N mineralization values for the bare, unamended soil (22.72 mg $NO_3$-N kg$^{-1}$ and 18.45 mg $NO_3$-N kg$^{-1}$, respectively), and it is important to note that the higher control values contribute to the narrowing of the range mineralization capacity in the later freeze and thaw periods. This increase observed in the bare, unamended soil may be due to a natural microbial shift as an adaptation to the fact that no new carbon source was added to the soil only controls [41]. In other words, microbes in the bare, unamended soil recognize no carbon source addition and start to consume SOM in those controls as an energy source whereby $NO_3$-N is being released.

When examining the five incubation cycles, the forage radish and pea (narrow C/N ratio) are the only crop residue treatments that show a significant increase in the soil $NO_3$-N mean from the control (Figure 1).

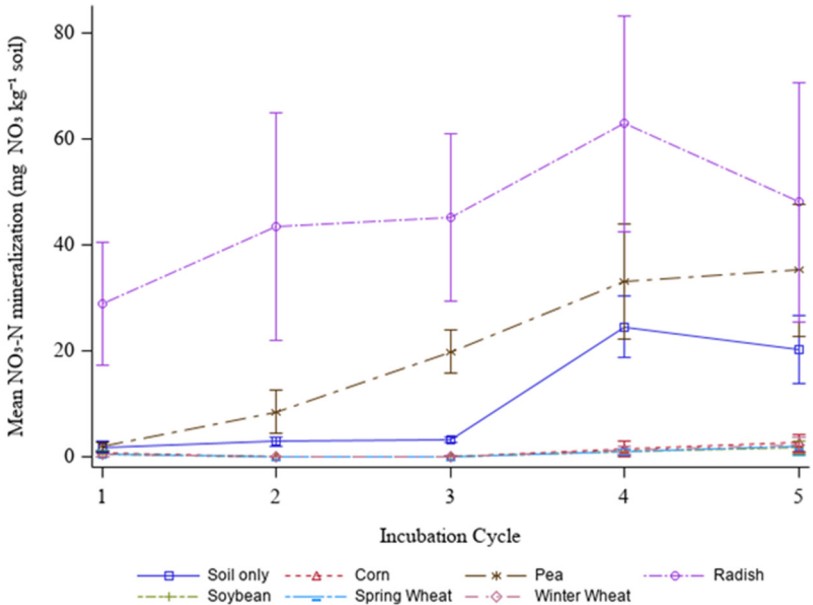

**Figure 1.** $NO_3$-N mineralization means and ranges in value for soil control and corn, pea, radish, soybean, spring wheat, and winter wheat crop residue treatment over five incubation cycles.

During cycle 1, the forage radish is the only crop residue treatment significantly different from the other crop residues treatment (Figure 2). In cycle 2, the soil $NO_3$-N mean increases in both the bare, unamended soil (control) and the pea crop residue treatment, although they are still significantly different from the forage radish crop residue treatment (Figure 2). For cycle 3, the forage radish crop residue treatment still remains significantly different from the control, although both are similar to the soil $NO_3$-N mean for the pea

crop residue treatment (Figure 2). In cycle 4 and cycle 5, the bare, unamended soil (control) and the pea crop residue treatment are not significantly different from the forage radish, nor were they significantly different from the corn, soybean, spring wheat, and winter wheat crop residue treatments (Figure 2). By cycle 5, it is evident that the corn, soybean, spring wheat, and winter wheat crop residue treatments were not mineralizing soil $NO_3$-N and were always significantly different from the forage radish crop residue treatment. A reason for the observations in these mineralization patterns is the C/N ratio for each crop residue treatment influences the N mineralization characteristic of the residue [26].

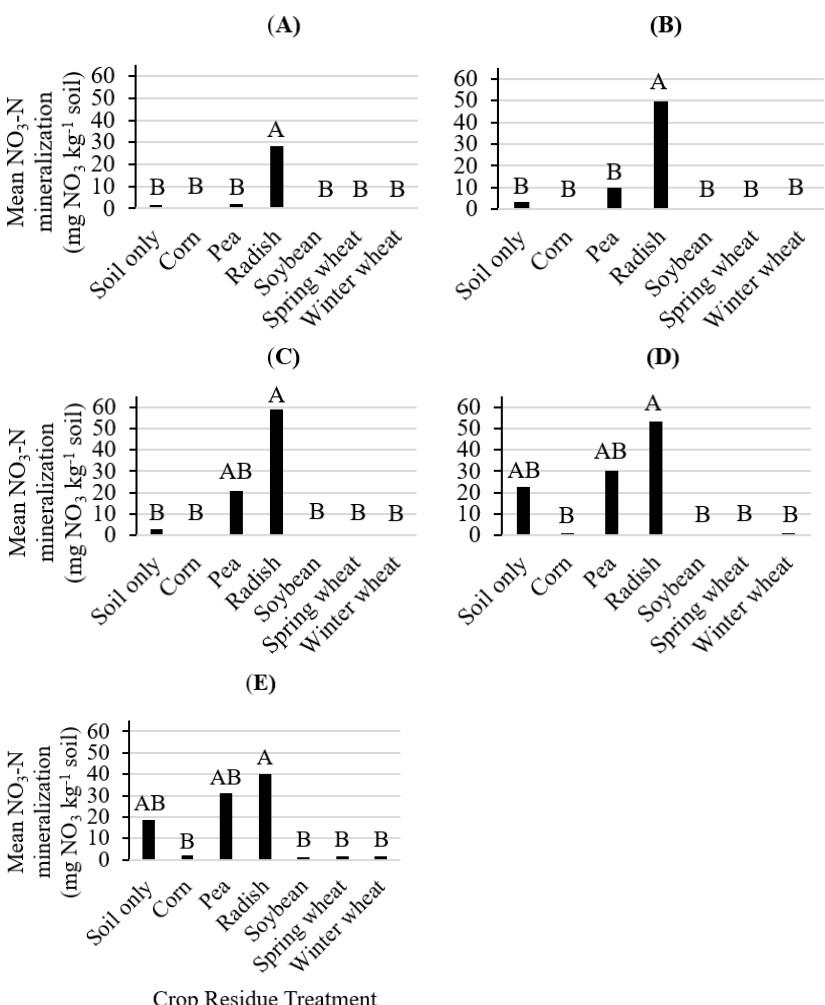

**Figure 2.** First incubation (**A**), second incubation (**B**), third incubation (**C**), fourth incubation (**D**), and fifth incubation (**E**) cycle mean soil $NO_3$-N mineralization patterns for control, corn, pea, radish, soybean, spring wheat, and winter wheat crop residue treatments. Different letters within each graph indicate significant difference at the 0.05 level using Tukey's comparison test.

For example, the forage radish (C/N = 8) $NO_3$-N mineralization mean is significantly higher when compared to each crop residue treatment and the bare, unamended soil during all five freeze and thaw periods. Pea (C/N = 18) shows increased N mineralization over the bare, unamended soil in all five freeze and thaw periods, with a steady increase in mineralization beginning in cycle 3 and beyond. All other crop residue treatments with higher C/N ratios (corn (C/N = 73), soybean (C/N = 53), spring wheat (C/N = 76), and winter wheat (C/N = 101)) exhibit patterns of soil $NO_3$-N immobilization.

A closer examination of the mean soil $NO_3$-N mineralization patterns over freeze and thaw periods (Figure 1) indicates a wide range in soil $NO_3$-N variation from the mean that is consistent for the forage radish crop residue treatment. This is because the forage radish

crop residue possesses available N and other nutrients required by microorganisms, in addition to what the soil harbors itself; therefore, the microbes have plentiful food options based upon their wants and needs and N contained in the plant material. Conversely, the wider C/N ratio crops (corn, soybean, spring wheat, and winter wheat) exhibit narrower soil $NO_3$-N variation from the mean N mineralization from the soil and possess limited nutrient availability in the soil, creating an environment where it is much harder for microbes to extract N, illustrating the inverse relationship with the C/N ratio (i.e., as the C/N ratio decreases, range increases). With this information alone, it appears that freeze and thaw effects on mean soil $NO_3$-N mineralization vary based on crop residue treatment and the C/N ratio of the material.

The bare, unamended soil, absent of any crop residue treatment, shows a cumulative soil $NO_3$-N mineralization pattern that increases with freeze and thaw periods. Cumulative values of soil $NO_3$-N mineralization of the five cycles are 11, 19, 19, 147, and 122 mg $NO_3$-N kg$^{-1}$, respectively (Table S1). From this, it is evident that soil $NO_3$-N mineralization in the control itself is increasing in the absence of any crop residue on the soil surface. This cumulative build-up may be evidence of the microbial shift mentioned previously that is preferentially attacking the native SOM. Because there is no carbon source added to the soil, microbes compete for a relatively stable (C/N = 10–12) nutrient source, whereby the N is then released from stable SOM. When examining soil $NO_3$-N mineralization further over the incubation cycles for each crop residue treatment, mineralization shifts from cycles representing the early growing season towards the mid-growing season cycles with increased freeze and thaw periods. For example, during the first incubation cycle, on day 14, soil $NO_3$-N mineralization values are significantly higher than the other incubation cycles (days) for each crop residue treatment (Figure 3). During the second incubation cycle, on day 28, soil $NO_3$-N mineralization is significantly higher for all crop residue treatments and day 14 shows the lowest soil $NO_3$-N mineralization values overall (Figure 3). In cycle 3, no significant differences occur among incubation cycles for all crop residue treatments (Figure 3). The shift becomes evident in the fourth incubation cycle, where soil $NO_3$-N mineralization is significantly different on days 42, 56, and 70 from the other incubation cycles (Figure 3). In the fifth incubation cycle, soil $NO_3$-N mineralization is significantly higher in the mid-growing season (day 42) from all other incubation cycles, while day 84 indicates significantly lower soil $NO_3$-N mineralization values (Figure 3). This pattern of $NO_3$-N mineralization release with the increasing number of freeze and thaw periods is of importance to pinpoint when nutrients might be available to plants. From this analysis, increased freeze and thaw periods can contribute to nutrient availability in the mid-growing season period, when plants often need it the most.

When evaluating the individual crop residue treatments and their contributions to soil $NO_3$-N mineralization averaged over all incubation cycles these can be ranked as forage radish > pea > bare, unamended soil $\geq$ corn = winter wheat = spring wheat = soybean (47.80 > 19.86 > 10.57 > 0.94 = 0.76 = 0.70 = 0.66 mg $NO_3$-N kg$^{-1}$, respectively). Forage radish is significantly different from all crop residues treatments and the bare, unamended soil. The pea crop residue treatment is significantly different from the bare, unamended soil. The corn, soybean, spring wheat, and winter wheat crop residue treatments are always less than mg $NO_3$-N kg$^{-1}$, are similar, and significantly different from the forage radish, pea, and unamended soil control.

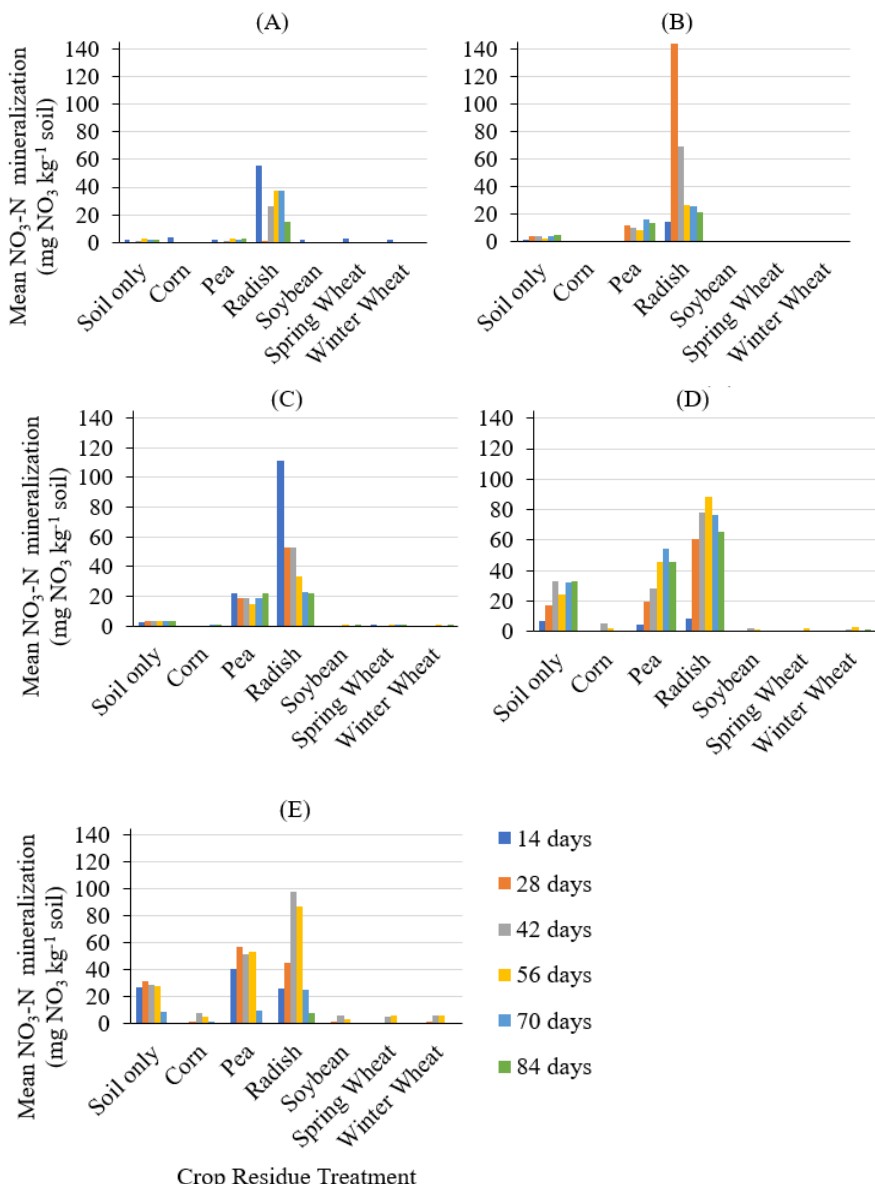

**Figure 3.** First incubation (**A**), second incubation (**B**), third incubation (**C**), fourth incubation (**D**), and fifth incubation (**E**) cycle mean soil NO$_3$-N mineralization patterns for control, corn, pea, radish, soybean, spring wheat, and winter wheat crop residue treatments by incubation period.

Figure 4 shows the average daily trends of soil NO$_3$-N mineralization and immobilization from the crop residue treatments during each incubation cycle (*n* = 5) to determine whether NO$_3$-N in the crop residue contributes to N availability in the soil. The bare, unamended soil is represented by the zero line. Above the zero line indicates N mineralization, while below the zero line suggests N immobilization. Mineralization/immobilization quantities differ based on the number of incubation cycles. Daily mineralization and immobilization NO$_3$-N values ranged from −0.20 to 2.53 mg NO$_3$-N kg$^{-1}$ for the first cycle, from −0.34 to 18.61 mg NO$_3$-N kg$^{-1}$ for the second cycle, from −0.30 to 7.74 mg NO$_3$-N kg$^{-1}$ for the third cycle, from −2.32 to 4.60 mg NO$_3$-N kg$^{-1}$ for the fourth cycle, and from −2.15 to 4.96 mg NO$_3$-N kg$^{-1}$ for the fifth cycle. From these figures, it is evident that forage radish and pea are the only crops mineralizing at or above the bare, unamended soil control levels. All other crops (corn, soybean, winter wheat, and spring wheat) show N immobilization for each leaching of every incubation cycle.

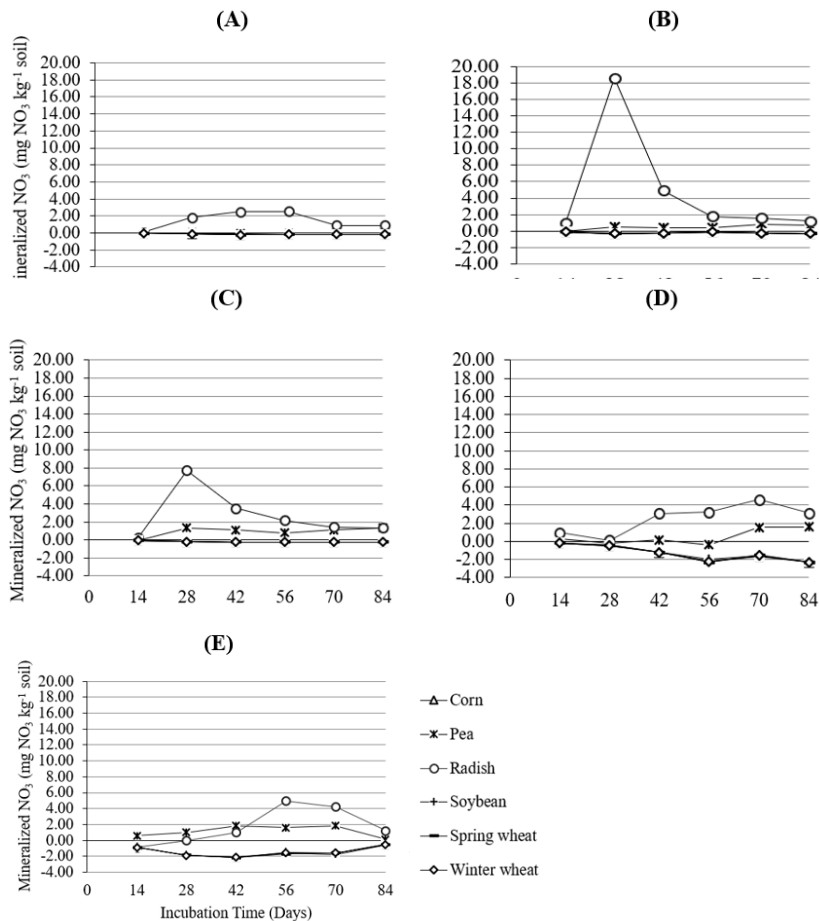

**Figure 4.** First incubation (**A**), second incubation (**B**), third incubation (**C**), fourth incubation (**D**), and fifth incubation (**E**) cycle daily mean mineralization/immobilization of soil $NO_3$-N.

These N mineralization/immobilization trends are consistent with the results provided earlier from crops with a narrower C/N ratio being the only ones to show mineralization above the bare, unamended soil. Patterns of the average daily $NO_3$-N mineralization/immobilization trends are also similar to the trends observed when examining the overall mean for each freeze and the cycle, where increases are observed in the earlier growing season for cycles 1 and 2 and where increases are observed in the mid to late growing season in cycles 4 and 5. These patterns are also generally consistent regardless if above or below the control, although they are heavily influenced by the forage radish results.

## 4. Conclusions

Accumulation of crop residue on the soil surface did not increase soil $NO_3$-N mineralization. Narrow C/N ratio crop residue treatments influenced the rate of soil $NO_3$-N mineralization, as was evident with the forage radish and pea crop residue treatments being the only ones to mineralize. This can help to inform freeze and thaw effects on $NO_3$-N mineralization in a frigid environment such as North Dakota, where producers are limited to short growing seasons. The number of incubation cycles may have had a direct effect on the mineralization process for all the crop residue treatments, where cycle 4 and cycle 5 were noted to release the most $NO_3$-N, particularly in the mid to late growing season as cycles increase. Wide C/N ratio crop residue such as corn, soybean, spring wheat, and winter wheat showed a net immobilization effect for every incubation cycle. The bare, unamended soil showed a cumulative increase in $NO_3$-N mineralization cycle-over-cycle. It would have been expected that the addition of a carbon source to the soil surface, in the form of crop residue, would have been at or above the rate of the control as carbon would act as a new nutrient source for microorganisms to decompose and release nutrients back to

the soil (e.g., nitrate). This was not found to be true and indicates a state where degradation of the native SOM occurs, which may eventually contribute to soil degradation. In other words, the absence of a carbon source addition may lead microorganisms to seek a carbon source from the soil itself. When this occurs, it is due to the lack of nutrient substrate from organic matter. Thus, the microorganisms themselves release nutrients, including nitrogen, from the organic matter in the bare, unamended soil. Future studies may consider examining the addition of nitrogen fertilizer to determine whether it may aid in facilitating soil $NO_3$-N mineralization in fields under a long-term no-till system with wide C/N ratio crop residues, offsetting the mineralization of N by decomposing crop residues or releasing N from the death of microorganisms.

There are several directions in future research that can be informed by our results. Further studies should examine the effect of the freeze and thaw cycle as well as the microbial count and community to better understand soil $NO_3$-N mineralization source and sink. Microbes require nitrogen as a nutrient source; thus, limited nutrients in a cropping system may be observed where the volume of crop residue and the recommended fertilizer additions or credits are not equitable. In addition, further studies incorporating increased fertilizer may provide insights on the impacts of wide C/N ratio crop residues on $NO_3$-N mineralization source and sink. A limitation of this study was the homogenous crop residue on the soil surface, whereas in a field setting, a mixed residue application (i.e., a sequence of different crops in a rotation) would more closely align with field conditions in a production environment.

**Supplementary Materials:** The following supporting information can be downloaded at: https://www.mdpi.com/article/10.3390/nitrogen3020011/s1, Table S1: Mean and cumulative $NO_3$-N mineralization for nine incubation cycles and overall mean for corn, flax, pea, radish, soybean, spring wheat, and winter wheat crop residues for Heimdal-Emrick, Fargo, and Forman soil series in North Dakota.

**Author Contributions:** Conceptualization, L.C. and R.S.A.; methodology, L.C. and R.S.A.; validation, L.C.; formal analysis, Q.W.; investigation, R.S.A.; resources, L.C.; data curation, R.S.A.; writing—original draft preparation, R.S.A.; writing—review and editing, R.S.A., L.C., and Q.W.; supervision, L.C.; funding acquisition, L.C. All authors have read and agreed to the published version of the manuscript.

**Funding:** This research was funded by the North Dakota State Board of Research and Extension Corn Committee (grant # 18-5-0090 and 19-11-0090), Soybean Committee (grant #19-6-0139), and Wheat Committee (grant # 19-38-0162) with matching funding from the North Dakota Corn Council (grant # 18-5-0090 and 19-11-0090), North Dakota Soybean Council (grant #19-06-0139), and North Dakota Wheat Commission (grant #19-38-0162). This work was conducted in conjunction with NDSU Hatch Project No. FARG008572 contributing to Multistate Research Project NC-1178 (Land Use and Management Practices Impacts on Soil Carbon and Relate Agroecosystems Services) of the National Institute of Food and Agriculture, U.S. Department of Agriculture.

**Institutional Review Board Statement:** Not applicable.

**Informed Consent Statement:** Not applicable.

**Data Availability Statement:** Data supporting reported results can be accessed by contacting rashadalgamdi@hotmail.com.

**Acknowledgments:** This research is part of Alghamdi's doctoral thesis under the supervision of Cihacek, and it was published at North Dakota State University database.

**Conflicts of Interest:** The authors declare no conflict of interest. The funders had no role in the design of the study; in the collection, analyses, or interpretation of data; in the writing of the manuscript, or in the decision to publish the results.

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
