# Peer review of "Simulated Cropping Season Effects on N Mineralization from Accumulated No-Till Crop Residues"

_nitrogen, doi:10.3390/nitrogen3020011_

Round 1

Reviewer 1 Report

This manuscript presents an incubation study of soil nitrogen (N) mineralization by adding crop residuals. Given that no-till or conservation tillage management practices is common in the North Dakota region, the authors wish to explore whether soil N supply is changed with surface covered by crop residuals. With a series of incubations, they found that crop residuals with narrow C/N would stimulate soil nitrate mineralization, while those with wide C/N even showed net N immobilization. In addition, they found enhanced soil N mineralization in the unamended soils, which may be attributed to freeze-thawing cycles or shift of soil microbial activity. Overall I found the results were interesting, but a major revision is necessary before it could be reconsidered for publication.

General comments:

In this manuscript, I found that the major findings were quite straight-forward, but the presentation style was in a complexed way. For example, in the introduction, literature reviewing of climatic factors and N fertilization on soil mineralization were not all necessary. In my opinion, it could be more concise, and the major hypothesis should be highlighted in the beginning.

It is unclear to me how mineralization rates were determined or defined in this study. As noted in the method, inorganic N was measured from the leachates, which likely represent net mineralization. However, in the field, gross N turnover should represent soil N supply capacities better. So this leaves a question to the authors: are these results obtained from current experimental design adequate to address your research question?     

According to Table S1, nitrogen mineralization was indeed measured over different time, which should provide information for the authors to determine process rates. However, these mineralization processes were presented in the unit of “mg kg-1”. Does it mean that they are qualitative results (“capacity”), or used for treatment comparison only? This should be clearly explained in sections of method and results.  

Detailed comments:

Line 95-96: “Result in…, resulting in”, quite redundant.

Line 159: Shouldn’t this be “five incubation cycles and four freeze-thaw periods” as referred to the figures.

Line 213-217: It is quite interesting here that nitrate mineralization rates increased during the last two incubation cycles. The authors tried to explain it: “may be due to a natural microbial shift as an adaptation to the fact that no new carbon source was added to the soil only controls.” Was there any reference? In addition, I can only find soil nitrate mineralization results throughout this work, while little is known for ammonium. Could the authors provide other measurement data to help explain their finding here?

Line 218-220: The C/N ratios of all applied residuals need to be provided in the manuscript or in a separate table.

Line 267: This should be Table S1 instead.

Figure 3: The black and white style makes it hard to distinguish treatments. Why not use colors?

Figure 4: N immobilization is not indicated among these five figures.

Author Response

General comments:

In this manuscript, I found that the major findings were quite straight-forward, but the presentation style was in a complexed way. For example, in the introduction, literature reviewing of climatic factors and N fertilization on soil mineralization were not all necessary. In my opinion, it could be more concise, and the major hypothesis should be highlighted in the beginning.

            It is our experience that much reported recent research has not accounted for changes in climactic variables (temperature and moisture that affect N availability, utilization and efficiency). For example, in many parts of the northern Great Plains, we have been through an approximately 24 year (1993-2017) period of precipitation higher than the normal for the previous 3-5 decades. However, since 2017, we are seeing what appears to be a return to the variability of the previous decades including extreme drought (2 of 5 years). Most current fertilizer recommendations are based on experiences of the wet period. In addition, improved, higher-yielding crop varieties also are producing higher residue yields and these are not always accounted for in recent N recommendation models. In our longer introduction we are trying to clearly lay out the argument that in cooler climates, greater attention needs to be given to effects of specific crops in current cropping systems and how processes could influence our N recommendations.

It is unclear to me how mineralization rates were determined or defined in this study. As noted in the method, inorganic N was measured from the leachates, which likely represent net mineralization. However, in the field, gross N turnover should represent soil N supply capacities better. So this leaves a question to the authors: are these results obtained from current experimental design adequate to address your research question?  ….

                We thank the author for their thoughtful question. Yes, the method of Stanford and Smith is a well established method that is often used to evaluate net N mineralization as NO3--N production. This study was designed to provide us with an idea of a baseline of what N mineralization should occur- under optimum field conditions (which are rarely optimum). Condition in the field will be characterized by highly variable moisture and temperature conditions including extreme dryness and waterlogging. Our research included an open incubation system (free aeration) and a constant temperature near the average seasonal soil temperature based on records from local weather stations. This research only partially answers the research question. We are following up with ongoing field research with continuous temperature and moisture monitoring to obtain further answers to this question.

According to Table S1, nitrogen mineralization was indeed measured over different time, which should provide information for the authors to determine process rates. However, these mineralization processes were presented in the unit of “mg kg-1”. Does it mean that they are qualitative results (“capacity”), or used for treatment comparison only? This should be clearly explained in sections of method and results. 

                These are quantities (quantitative) that can be used to compare treatments and mineralization rates (kinetics). These are units that make sense to researchers and farmers because these values are adjusted on a unit of soil basis which can be then translated to kg ha-1 and relatable to nutrient rates farmers are working with. We have added wording in the Materials and Methods section to clarify this (lines 193-194).

Detailed comments:

Line 95-96: “Result in…, resulting in”, quite redundant.

The authors agree. This has been updated.

Line 159: Shouldn’t this be “five incubation cycles and four freeze-thaw periods” as referred to the figures.

We appreciate this comment and reviewed and updated this to be consistent throughout the manuscript.

Line 213-217: It is quite interesting here that nitrate mineralization rates increased during the last two incubation cycles. The authors tried to explain it: “may be due to a natural microbial shift as an adaptation to the fact that no new carbon source was added to the soil only controls.” Was there any reference? In addition, I can only find soil nitrate mineralization results throughout this work, while little is known for ammonium. Could the authors provide other measurement data to help explain their finding here?

  1. We thank the reviewer for the question and we have added the citation (line 234).
  2. The ammonium results for the leaching study were not included because soil ammonium concentrations are generally relatively constant, and if available it is subjected to nitrification or volatilization in our soils. We did measure the ammonium in our study and the readings were mostly not significant except in the first incubation cycle for the forage Radish but not significant differences were found from other treatments during the five incubation cycles as shown in the table below. This is due to a characteristic rapid ammonification by the radish material early in the mineralization (decomposition) process.

Line 218-220: The C/N ratios of all applied residuals need to be provided in the manuscript or in a separate table.

Done and the table has been added as Table 1 in the manuscript lines 139-140.

Line 267: This should be Table S1 instead.

Done.

Reviewer 2 Report

Comments on the manuscript entitled “Simulated cropping season effects on N mineralization from 2 accumulated no-till crop residues”

Overall
The topic of the manuscript is of high interest of international readership. The research on no-till management practices of croplands is highly important issue. They are of key problem in when we observe harsh environmental conditions: short growing season, low soil temperature, delayed soil warming and drying. The above-mentioned conditions influence the processes of microorganisms' activity and thus, the mineralization of nitrogen and its retention in the soil.

  • The work is well planned and logical. The results are well described.
  • The graphical presentation of the findings needs minor improvements in the quality of the figures e.g., Fig. 3.
  • The results and their conclusions are helpful for no-till cultivation under similar climatic conditions.
  • Based on the conducted research, the Authors of the reviewed paper have made some new hypotheses, which they would like to verify.

Author Response

The work is well planned and logical. The results are well described.

The graphical presentation of the findings needs minor improvements in the quality of the figures e.g., Fig. 3.

We appreciate the reviewer’s comment and we have revised the figure for clarity 

The results and their conclusions are helpful for no-till cultivation under similar climatic conditions.

Based on the conducted research, the Authors of the reviewed paper have made some new hypotheses, which they would like to verify.

Round 2

Reviewer 1 Report

I have reviewed an earlier version of this manuscript. According to the revision and the authors' reply, I am happy that most of my concerns have been addressed.

Therefore, I recommend it for publication as it is at this stage.